# Adolescent betting on esports using cash and skins: Links with gaming, monetary gambling, and problematic gambling

Nerilee Hing[1]*, Lisa Lole[1], Alex M. T. Russell[1‡], Matthew Rockloff[1‡], Daniel L. King[2‡], Matthew Browne[1‡], Philip Newall[1‡], Nancy Greer[1‡]

1 Experimental Gambling Research Laboratory, School of Health, Medical and Applied Sciences, CQ University, Bundaberg, QLD, Australia, 2 College of Education, Psychology, and Social Work, Flinders University, Adelaide, SA, Australia

☯ These authors contributed equally to this work.
‡ These authors also contributed equally to this work
* n.hing@cqu.edu.au

**Data Availability Statement:** Data are owned by the New South Wales Office of Responsible Gambling (https://www.responsiblegambling.nsw. gov.au). To access these data, others would need

## Abstract

Adolescents can easily access esports betting sites and place bets using cash or skins. This descriptive cross-sectional study examined the characteristics of adolescent esports bettors and relationships between their esports betting, video gaming activities, monetary gambling participation, and at-risk/problem gambling. Two survey samples of Australians aged 12–17 years were recruited through advertisements ($n = 841$) and online panel providers ($n = 826$). In both samples, gender and parents' living situation did not differ by past-month esports cash and skin betting, but recent esports betting was associated with engaging in esports gaming activities such as playing and watching esports, and in monetary gambling activities. Past-month esports betting using cash and skins was significantly associated with at-risk/problem gambling. After controlling for recent monetary gambling, recent esports skin bettors were over 3 times more likely to meet criteria for at-risk/problem gambling. Esports betting using skins appears to pose risks for young people and is easily accessible through unlicensed operators.

## Introduction

Esports events are professionally organised video game competitions between players or teams who compete in first-person shooter, sports, action, strategy, or collectible card games [1,2]. Esports competitions transform recreational video gaming into professional gameplay, as players compete to win prize money and/or a championship title [3,4]. Esports has attracted a strong community of fans who watch the competitions online on dedicated television channels, streams, or at live events, and interact in online forums [2,4]. It has been estimated that, globally, there will be 728.8 million live-streaming esports viewers in 2021, and US$1.1 billion in esports revenues which would be a 14.5% increase since 2020 [3]. Watching esports events is particularly popular among adolescents who also report higher involvement than adults in recreational playing of esports games [5–7].

to first obtain approval to access the data from the New South Wales Office of Responsible Gambling. Phone +61 2 9995 0992 or Email: info@responsiblegambling.nsw.gov.au. Once approval is granted, the New South Wales Office of Responsible Gambling would liaise with the authors to ensure the relevant dataset is provided to the requesting party.

**Funding:** NH, AR, MR, MB, PN, NG and DK received grant funding to conduct this research (no grant number assigned). Funding for this study was provided by the NSW Government's Responsible Gambling Fund, with support from the NSW Office of Responsible Gambling https://www.responsiblegambling.nsw.gov.au. The funders reviewed the survey instrument for this study but had no role in data collection, analysis, decision to publish, or preparation of the manuscript.

**Competing interests:** The authors have declared that no competing interests exist.

Esports betting can involve placing bets using cash, or alternative currencies including monetised 'skins.' Esports cash betting is offered by established wagering operators and newer esports betting-exclusive operators [1]. These operators may or may not be licensed to provide gambling services but are nevertheless easily accessed online. Licensed operators usually have age verification protocols to prohibit minors; however, unregulated operators may not be as diligent. In contrast, esports skin gambling websites allow skins obtained in some video games to be wagered and rarely have any age controls. Skins are cosmetic virtual in-game items that can enhance and customise a player's avatar, weapons or equipment in a video game, although these skins do not usually affect game-play performance [8,9]. Skins, nonetheless, have monetary value due to aesthetic appeal and relative rarity, and are used as a currency for betting on esports. Because skin gambling operators are poorly monitored by most jurisdictions and have no, or weak, age verification, esports skins betting can be easily accessed by adolescents [8,10].

Gambling is a popular activity amongst adolescents [11], even though those aged under 18 years are below the legal gambling age in many countries, including Australia. Nonetheless, gambling problems are at least as prevalent, if not more so, amongst adolescents than adults [12]. Video games are, likewise, immensely popular in this demographic and evidence has shown that that video games may provide a 'gateway' to gambling [1,13–16]. Several researchers have expressed concerns about underage betting on esports [1,17–20] but there is little empirical research on this topic. Two international studies involving convenience samples of youth, 27% of whom were under the legal gambling age, reported associations between esports spectating and esports betting, as well as participation in other gambling activities and gambling problems [18,21]. Separate analyses were not included for the underage cohorts, and further research is needed to determine whether young people's participation in esports activities increases their risk of gambling harm.

In the UK, a survey of emerging adults, aged 16–25 years, weighted by gender and region, assessed lifetime engagement in esports betting. Amongst the 962 respondents, 7% reported betting on esports prior to age the legal gambling age of 18, with 5% of those aged 16–17 years betting on esports in the previous four weeks [22]. However, the study did not distinguish whether this esports betting was with cash, skins or both. In a non-probability panel sample of 3,549 UK residents aged 16–24 years, 1.8% of those aged under 18 years reported past-year cash betting on esports, but esports skin betting was subsumed into a broader measure of gambling-with-skins on external sites [23]. In the whole sample, and compared to sports bettors and non-gamblers, esports cash bettors were more likely to be male, from non-white ethnic groups, heavily involved in playing digital games, and to have higher rates of gambling involvement and problem gambling [23]. A study in New South Wales (NSW), Australia, also reported low rates of esports cash betting (1.4%) amongst 551 adolescents aged 12–17 years [11]. This sample was recruited through flyers for delivery to all in-scope households in NSW, but COVID-19 lockdowns and flyer delivery problems may have affected the sample's representativeness. Overall, studies to date indicate modest rates of esports cash betting amongst adolescents and emerging adults.

Esports skin betting appears to be a larger market, although market size has varied with periodic actions taken to restrict skin betting websites [1]. Only a few studies have examined skin gambling among adolescents, and most have not separately assessed participation in esports skin gambling and use of skins from other gambling activities, such as on chance-based games, other competitive events, and private betting [9,23,24–26]. Hing et al.'s [11] study appears to be the only research to have distinguished between esports skin betting and skin gambling on other activities. It found that 6.2% of adolescents under the legal gambling age of 18 years reported esports skin betting in the past year, which was 2.5 times higher than for esports cash betting.

A key issue is whether esports betting is linked to gambling problems amongst young people. However, research into adolescent esports betting and gambling problems has been limited to cross-sectional associations in mixed adult-adolescent samples [21,23]. These studies, and research with adults, have found that greater frequency of esports betting is associated with greater problem gambling severity [13,27–30]. The relationship between esports betting and gambling problems is, however, less clear when controlling for the frequency of other gambling activities. A representative adult prevalence study in Victoria, Australia, found that esports cash betting was the gambling activity most strongly associated with gambling harm, when controlling for engagement in other monetary gambling activities [29]. However, another Australian study with a convenience sample found that the relationship between esports cash betting and gambling problems was no longer significant when controlling for other gambling forms [28]. In an Australian panel sample of 298 regular esports bettors aged 18 years or over [13], esports skin betting, but not esports cash betting, significantly predicted problem gambling severity while controlling for gambling on other forms.

Overall, little is known about the characteristics of adolescents who bet on esports with cash and with skins. While esports betting appears to be associated with gambling problems, whether adolescent esports bettors engage in other gambling forms that may, instead, be the source of such problems, has not been considered. This study therefore aims to address the following objectives relating to adolescents aged 12–17 years:

1. Examine demographic, psychological and video gaming characteristics of past-month esports cash bettors and esports skin bettors to see if they differ.

2. Examine adolescents' participation in other forms of gambling among past-month esports cash bettors and esports skin bettors.

3. Assess whether problem gambling severity varies according to past-month engagement in esports cash betting and esports skin betting.

4. Assess whether past-month esports cash betting and esports skin betting are associated with problem gambling severity when controlling for engagement in other monetary forms of gambling.

## Method

### Participants

A total of 1,667 residents in NSW Australia (62.2% male), aged 12–17 years, responded to an online survey. Participants were recruited via an online panel aggregator (Qualtrics sample, $n = 826$) and via advertisements (Advertisements sample, $n = 841$). These advertisements were placed on Facebook, Instagram and Twitter, in the funding agency's online communications (e.g., newsletters), and emailed to NSW participants in our laboratory's previous gambling studies who had consented to be re-contacted. These previous research participants were asked to invite any adolescents in their household to complete the survey.

In the Qualtrics sample, 4,101 people started the survey but 3,003 did not meet the eligibility criteria of parental consent ($n = 2,364$), own consent ($n = 119$) and age and location (residing in NSW) criteria ($n = 520$). A further 42 responses had data integrity issues, such as completing the survey too quickly, failing an attention check or poor-quality data. Of the remaining eligible 1,056 participants, 826 completed the survey for a completion rate of 78.2%. A total of 1,473 started the Advertisements survey but 69 did not meet the eligibility criteria of informed consent ($n = 9$) or age and location criteria ($n = 60$). Of the remaining 1,404 participants, 841 completed the survey, for a completion rate of 60.0%.

**Table 1. Key demographic, psychological and gaming characteristics of participants.**

| Variables | Advertisement Sample (*n* = 841) | | Qualtrics Sample (*n* = 826) | |
|---|---|---|---|---|
| | *n or M (SD)* | % | *n or M (SD)* | % |
| **Age** | 14.6 (1.7) | - | 14.8 (1.6) | - |
| **Gender** | | | | |
| Female | 258 | 30.7 | 370 | 44.8 |
| Male | 582 | 69.2 | 455 | 55.1 |
| Other | 1 | 0.1 | 1 | 0.1 |
| **Parent's Living Arrangements** | | | | |
| Living together | 532 | 63.3 | 637 | 77.1 |
| Not living together | 309 | 36.7 | 189 | 22.9 |
| **Wellbeing** | 7.21 (3.0) | - | 8.7 (1.9) | - |
| **Impulsiveness** | 19.0 (4.0) | - | 17.7 (4.4) | - |
| **Problem Gambling Severity** | | | | |
| Non-gambler | 261 | 31.0 | 419 | 50.7 |
| Non-problem gambler | 92 | 10.9 | 203 | 24.6 |
| At-risk gambler | 69 | 8.2 | 76 | 9.2 |
| Problem gambler | 419 | 49.9 | 128 | 15.5 |
| **Problematic Gaming Symptom Score** | 3.7 (2.3) | - | 2.3 (2.7) | - |
| **Past-month esports cash betting** | | | | |
| Yes | 289 | 34.4 | 81 | 9.8 |
| No | 552 | 65.6 | 745 | 90.2 |
| **Past-month esports skin betting** | | | | |
| Yes | 214 | 25.4 | 115 | 13.9 |
| No | 627 | 74.6 | 711 | 86.1 |

*Note.* The following inventories were used for each variable: *Wellbeing* = Personal Wellbeing Index–School Children [33]; *Impulsiveness* = Barratt Impulsiveness Scale–Brief (BIS-B) [34]; *Problem Gambling Severity* = DSM-IV-MR-J [31]; *Problematic gaming symptoms* = Internet Gaming Disorder [IGD, 32].

Due to significant differences between the samples on key variables, the descriptive summaries (Table 1) and inferential analyses (Tables 2 to 8) were conducted separately for these two samples. Conducting separate analyses also provided greater confidence in the results that were consistent across both samples.

## Materials

Participants completed a 15-minute online survey with the following measures.

### Esports cash betting

Respondents were asked '*When did YOU last. . . bet on esports events FOR REAL MONEY, like CS-GO, League of Legends or DOTA2*?' on a 5-item response scale. Response options '*In the last 7 days'* and '*In the last 4 weeks'* were re-coded as 'past-month esports cash betting'. '*In the last 12 months,'* '*More than 12 months ago,'* and '*Never'* were recoded as 'no past-month esports cash betting.'

### Esports skin betting

Respondents were asked '*When, if ever, did you last use in-game items for betting on the outcome of a competitive video gaming contest (esports betting).'* The same response options and recoding were used as above to delineate 'past-month esports skin betting' and 'no past-month esports skin betting.'

**Table 2. Multivariate logistic regression with demographic and psychological variables as predictors of past-month esports cash betting.**

| Variables | Advertisements Esports cash betting vs not (ref. = not) | | | | | Qualtrics Esports cash betting vs not (ref. = not) | | | | |
|---|---|---|---|---|---|---|---|---|---|---|
| | B | S.E. | Z | P | Odds Ratio [95% CI] | B | S.E. | Z | p | Odds Ratio [95% CI] |
| Age (in years) | .04 | .05 | .086 | .392 | 1.04 [.95,1.15] | .13 | .08 | 1.53 | .125 | 1.14 [.97,1.34] |
| Gender | -.17 | .18 | -.92 | .359 | .85 [.59,1.21] | -.03 | .27 | -.11 | .914 | .97 [.57,1.64] |
| ATSI | 1.10 | .17 | 6.49 | < .001 | 3.02 [2.16,4.21] | .66 | .34 | 1.98 | .048 | 1.94 [1.01,3.75] |
| Parents living together | .26 | .17 | 1.53 | .127 | 1.29 [.93,1.80] | .26 | .30 | .87 | .386 | 1.30 [.72,2.34] |
| Wellbeing | -.12 | .03 | -4.34 | < .001 | .88 [.84,.93] | -.08 | .07 | -1.23 | .220 | .92 [.81,1.05] |
| Impulsiveness | .08 | .02 | 3.44 | < .001 | 1.08 [1.03,1.13] | -.07 | .03 | -2.27 | .023 | .93 [.87,.99] |
| Problematic gaming | .18 | .17 | 1.05 | .293 | 1.20 [.86,1.67] | .53 | .28 | 1.89 | .059 | 1.70 [.98,2.95] |
| Played esports game | .21 | .17 | 1.25 | .213 | 1.24 [.89,1.72] | 1.28 | .31 | 4.19 | < .001 | 3.60 [1.98,6.56] |
| Watched esports | .81 | .17 | 4.77 | < .001 | 2.25 [1.61,3.13] | .39 | .31 | 1.25 | .212 | 1.48 [.80,2.74] |
| Competed in esports | -.14 | .36 | -.39 | .700 | .87 [.43,1.77] | .62 | .34 | 1.82 | .069 | 1.87 [.95,3.65] |
| Intercept | -3.12 | .93 | -3.35 | < .001 | .04 [.01,.27] | -3.20 | 1.57 | -2.04 | .041 | .04 [.002,.88] |

Missing data, indicated by dashed lines, are due to insufficient sample size for analyses ($n < 10$). ATSI = *Aboriginal and/or Torres Strait Islander*. Reference groups = No for *Parents Living Together*, *Paid Job*, *Internet Gaming Disorder*, *Played esports game*, *Watched esports*, and *Competed in esports*. Other reference groups are *Gender* = Female and *ATSI* = non-Indigenous.

Advertisements sample model fit: AIC = 920; $R^2_{McF}$ = .17. Qualtrics sample model fit: AIC = 427; $R^2_{McF}$ = .15.

### Frequency of participation in 10 other gambling activities

Participants were asked how often they used real money in the past month to gamble. This was measured using the same response scale and coding method described above (please see Table 4).

### Problem gambling symptoms

Problem gambling symptomology was assessed using the validated DSM-IV-MR-J [31]. The 9 items ask about addiction symptoms (e.g., '*In the past 12 months how often have you found yourself thinking about gambling or planning to gamble*?'), with a variety of response scales that are recoded into 'yes' or 'no'. Four or more 'yes' responses indicate problem gambling, and 2–3 'yes' responses indicate at-risk gambling. Cronbach's alpha for this questionnaire was .88 for the Qualtrics sample and .72 for the Advertisements sample.

### Video gaming behaviours

Respondents were asked if and when they last played an esports video game, watched an esports competition, and competed in an esports competition. '*In the last 7 days*' and '*In the last 4 weeks*' were re-coded as 'yes'. '*In the last 12 months*', '*More than 12 months ago*', and '*Never*' were recorded as 'no.'

### Problematic gaming symptoms

The Internet Gaming Disorder (IGD) scale [32] was used to assess problem video gaming symptomology. Response options were yes/no to 9 items relating to the past 12 months. Scores greater than 4 indicate problematic video gaming, as long as one of the endorsed items is: '*Did you risk or lose significant relationships, or job, educational or career opportunities because of gaming*?' Cronbach's alpha for this questionnaire was .84 for the Qualtrics sample and .60 for the Advertisements sample.

## Wellbeing

The validated *Personal Wellbeing Index–School Children* questionnaire [33] was used to measure participants' general wellbeing. This one item measure asks participants, '*How happy are you with your life as a whole*?' with a rating scale from 0 = *very sad* to 10 = *very happy*.

## Impulsivity

The 9-item *Barratt Impulsiveness Scale–Brief* (BIS-B) [34] was used to measure trait impulsivity. Response options are: 1 = *rarely/never*, 2 = *occasionally*, 3 = *often*, and 4 = *always* to questions such as, '*I act on the spur of the moment*.' Items 1, 4, 5, and 6 (e.g., '*I plan tasks carefully*') are reverse scored, then items summed to produce a total score out of 36. Cronbach's alpha for this questionnaire was .81 for the Qualtrics sample and .60 for the Advertisements sample.

## Demographic variables

These included age (in years), gender (male, female, other), identifying as of Aboriginal and/or Torres Strait Islander (ATSI) descent (yes or no), and parental living situation (parents living together vs not).

## Procedure

Respondents were able to proceed to the survey if they: indicated they were aged between 12 and 17 years; resided in NSW; had permission of a parent/guardian to participate; and they provided their consent. Participants were reimbursed via the panel provider's rewards system (for the Qualtrics sample) or with the option to enter a draw to win an AU $100 shopping voucher (for the Advertisements sample). The survey was conducted over three weeks in April-May 2020.

The study procedures were carried out in accordance with the Declaration of Helsinki. The study's protocol was approved by the Central Queensland University Human Research Ethics Committee. All subjects were informed about the study, and all provided written informed consent. Parental consent was also sought for all participants.

## Statistical analyses

A series of regression analyses were used to address the research objectives. The required sample size was determined by a power analysis (in the program G*Power; [35]) and feasibility estimates provided by the panel provider. Data from the Advertisements and Qualtrics samples were analysed separately to remain consistent with previous research and to ensure data transparency [11]. Covariates were not included in any of the analyses.

In order to examine demographic, psychological and video gaming characteristics of past-month esports cash bettors and esports skin bettors to see if they differ (Objective 1), a series of multivariate binary logistic regression analyses was used. These two analyses explored the degree to which these characteristics predicted the outcome variables, past-month engagement in esports cash betting (yes/no) and esports skin betting (yes/no). Continuous predictor variables in the model included age, personal wellbeing, and impulsiveness and the binary categorical variables of gender (male/female; non-binary gender was excluded from further analyses due to small sample sizes), Aboriginal and/or Torres Strait Islander (ATSI) descent (yes/no), parents' living situation (living together vs not), and problematic video gaming (IGD scores were used to group participants as yes/no, according to the recommended cut-off score of 5). Past-month gaming behaviours, including having played an esports game, watched an esports

competition, and/or competed in an esports competition, were also included as binary categorical predictors (yes/no) for each analysis.

To examine adolescents' participation in other forms of gambling among past-month esports cash bettors and esports skin bettors (Objective 2), separate multivariate binary logistic regressions were conducted. Past-month participation in 10 traditional gambling forms (yes/no) were treated as categorical predictor variables and past-month esports cash betting and esports skin betting (yes/no) were treated as the outcome variables (yes/no) in the respective analyses.

Binary logistic regression was used to assess whether problem gambling severity varies according to past-month engagement in esports cash betting and esports skin betting (Objective 3). This analysis coded the predictor variables, participation in esports cash betting and esports skin betting, as yes/no for the past-month participation. The outcome variable was whether participants reported at-risk/problem gambling or not. For this analysis, 'non-problem gamblers' were those participants who had gambled on at least one of the 10 gambling forms, but who scored zero on the DSM problem gambling criteria.

Finally, in order to assess whether past-month esports cash betting and esports skin betting are associated with problem gambling severity when controlling for engagement in other monetary forms of gambling (Objective 4), hierarchical logistic regressions were run. This analysis included past-month participation in the 10 monetary gambling forms (yes/no) in Block 1 for both samples. For the Qualtrics sample, past-month esports cash betting and esports skin betting participation (yes/no) were added as Blocks 2 and 3, respectively. Due to low numbers of non-problem past-month esports cash bettors, only esports cash betting (yes/no) was added as Block 2 for the Qualtrics sample analysis. Follow-up hierarchical logistic regressions were conducted, with the summed total number of activities engaged in (out of the 10 gambling forms), instead, being added into Block 1.

## Results

### Demographic, psychological, and gaming characteristics of esports cash bettors and esports skin bettors

The binary logistic regression to address Objective 1 revealed that few demographic and psychological characteristics predicted the two forms of betting. Gaming characteristics (playing and watching esports games) were found to be a stronger predictor of participation in such activities (especially skin betting), but results were not always consistent across the samples. For instance, past-month esports cash bettors in the Advertisements sample were at least twice as likely to have watched an esports competition in the past month, and were significantly more likely to report higher impulsivity and lower well-being than those who had not engaged in esports cash betting in the past month (Table 2). In the Qualtrics sample, past-month esports cash bettors were significantly more likely to have played an esports video game in the past month. In contrast to the Advertisements sample, Qualtrics participants in this sub-group reported lower impulsivity than the non-bettor comparison group. Being of Aboriginal and/or Torres Strait Islander descent was associated with higher odds of esports cash betting in both samples.

Past-month esports skin bettors in the Advertisements sample were more likely to report higher impulsivity, as well as having recently played an esports video game, watched esports competitions, and competed in an esports competition, compared to those who had not recently engaged in esports skin betting (Table 3). Past-month esports skin bettors in the Qualtrics sample were more likely to be older and to report problematic gaming symptoms, and also to have played, watched, and competed in esports, compared to non-recent bettors and non-bettors. Identifying as an Aboriginal and/or Torres Strait Islander person was associated

**Table 3. Multivariate logistic regression results with demographic and psychological variables as predictors of past-month skin betting.**

| Variables | Advertisements Esports skin betting vs not (ref. = not) | | | | | Qualtrics Esports skin betting vs not (ref. = not) | | | | |
|---|---|---|---|---|---|---|---|---|---|---|
| | B | S.E. | Z | p | Odds Ratio [95% CI] | B | S.E. | Z | p | Odds Ratio [95% CI] |
| Age (in years) | -.05 | .05 | -.99 | .324 | .95 [.86,1.05] | .18 | .08 | 2.35 | .020 | 1.20 [1.03,1.40] |
| Gender | -.17 | .19 | -.92 | .359 | .84 [.58,1.22] | .30 | .26 | 1.17 | .240 | 1.35 [.82,2.23] |
| ATSI | .78 | .18 | 4.25 | < .001 | 2.19 [1.52,3.14] | .53 | .33 | 1.57 | .116 | 1.69 [.88,3.26] |
| Parents living together | .27 | .18 | 1.57 | .117 | 1.32 [.93,1.87] | .06 | .29 | .20 | .844 | 1.06 [.61,1.85] |
| Wellbeing | -.05 | .03 | -1.63 | .104 | .95 [.90,1.01] | -.09 | .06 | -1.41 | .157 | .92 [.81,1.04] |
| Impulsiveness | .07 | .02 | 3.06 | .002 | 1.07 [1.03,1.13] | -.01 | .03 | -.48 | .631 | .99 [.93,1.05] |
| Problematic gaming | .30 | .18 | 1.70 | .089 | 1.35 [.95,1.91] | .79 | .25 | 3.11 | .002 | 2.21 [1.34,3.63] |
| Played esports game | .43 | .18 | 2.39 | .017 | 1.53 [1.08,2.18] | 1.09 | .28 | 3.93 | < .001 | 2.96 [1.72,5.09] |
| Watched esports | .66 | .18 | 3.65 | < .001 | 1.94 [1.36,2.78] | .62 | .28 | 2.21 | .027 | 1.86 [1.07,3.24] |
| Competed in esports | 1.53 | .35 | 4.47 | < .001 | 4.60 [2.36,8.98] | 1.61 | .31 | 5.14 | < .001 | 5.01 [2.71,9.27] |
| Intercept | -2.65 | .96 | -2.76 | .006 | .07 [.01,0.46] | -5.00 | 1.49 | -3.35 | < .001 | .01 [.004,.13] |

ATSI = *Aboriginal and/or Torres Strait Islander*. *Age*, *Wellbeing*, and *Impulsiveness* were measured on a continuous scale. All other variables were treated as categorical with the reference groups = No for *Parents Living Together*, *Paid Job*, *Internet Gaming Disorder*, *Played esports*, *Watched esports*, and *Competed in esports*. Other reference groups are *Gender* = Female and *Indigenous* = Non-Indigenous. Advertisements sample model fit: AIC = 854; $R^2_{McF}$ = .17. Qualtrics sample model fit: AIC = 520; $R^2_{McF}$ = .25.

with increased odds of esports skins betting in the Advertisements, but not the Qualtrics, sample.

## Esports cash bettors' and esports skin bettors' participation in other gambling activities

The binary logistic regression used to address Objective 2 revealed that participation in most other forms of gambling (especially for participants in the Advertisements sample) predicted involvement in esports cash and skin betting. In the Advertisements sample, recent esports cash betting was associated with a greater likelihood of recently gambling on EGMs, races, scratchies/lotto/pools, bingo/housie, fantasy sports, and informal private betting (Table 4). Due to low numbers in some cells, between-group comparisons could not be made for Keno, poker, casino games, and sports. In the Qualtrics sample, past-month esports cash bettors were also more likely to report recent gambling on lotto/pools/scratch lotto tickets, bingo, fantasy sports, and informal private betting.

Participants in the Advertisements sample who reported esports skin betting in the past month were more likely to have recently gambled on racing, lotto/pools/scratch lotto tickets, bingo/housie, and informal private betting, compared to non-recent bettors and non-bettors. Due to low numbers in some cells, between-group comparisons could not be made for Keno, poker, casino games, and sports (Table 5). In the Qualtrics sample, recent esports skin betting was associated with recent gambling on lotto/pools/scratch lotto tickets, bingo/housie, and informal private betting, compared to those who had not recently bet on esports with skins.

## The relationship between esports cash betting and esports skin betting and at-risk/problem gambling

Esports skin betting significantly predicted at-risk/problem gambling symptomology in both samples. There were not enough non-problem esports cash bettors for analysis to address Objective 3 in the Advertisements sample, but, as noted, the relationship between esports skin

**Table 4. Multivariate logistic regression results with past-month participation in monetary gambling activities as predictors of past-month esports cash betting.**

| | Advertisements Esports cash betting vs not (ref. = not) | | | | | Qualtrics Esports cash betting vs not (ref. = not) | | | | |
|---|---|---|---|---|---|---|---|---|---|---|
| Variables | B | S.E. | Z | P | Odds Ratio [95% CI] | B | S.E. | Z | P | Odds Ratio [95% CI] |
| EGMs | .69 | .18 | 3.79 | < .001 | 2.00 [1.40,2.86] | .57 | .53 | 1.08 | .280 | .02 [.01,.03] |
| Racing | 1.05 | .41 | 2.58 | .010 | 2.87 [1.29,6.38] | -.88 | .59 | -1.50 | .133 | .41 [.13,1.31] |
| Scratchies, lotto, pools | .67 | .19 | 3.59 | < .001 | 1.95 [1.35,2.80] | 1.22 | .36 | 3.44 | < .001 | 3.40 [1.69,6.82] |
| Keno | - | - | - | - | - | .62 | .52 | 1.20 | .229 | 1.87 [.68,5.16] |
| Bingo or housie | .63 | .19 | 3.27 | .001 | 1.88 [1.29,2.74] | 1.24 | .41 | 3.01 | .003 | 3.44 [1.54,7.71] |
| Poker | - | - | - | - | - | .14 | .61 | .23 | .820 | 1.15 [.35,3.76] |
| Casino games | - | - | - | - | - | 1.26 | .65 | 1.93 | .054 | 3.51 [.99,12.57] |
| Sports | - | - | - | - | - | -.15 | .52 | -0.28 | .778 | .86 [.31,2.40] |
| Fantasy sports | 1.02 | .19 | 5.49 | < .001 | 2.78 [1.93,4.01] | 2.16 | .45 | 4.84 | < .001 | 8.67 [3.62,20.78] |
| Informal private betting | 1.00 | .18 | 5.51 | < .001 | 2.71 [1.90,3.86] | 1.53 | .36 | 4.30 | < .001 | 4.61 [2.30,9.27] |
| Intercept | -2.27 | .16 | -14.20 | < .001 | .10 [.08,0.14] | -3.98 | .27 | -14.91 | < .001 | .02 [.01,.03] |

Missing data, indicated by dashed lines, are due to insufficient sample size for analyses ($n < 10$). *EGMs* = electronic gaming machines.

The reference group for all variables = No (did not participate in the past month).

Advertisements sample model fit: AIC = 844; $R^2_{McF}$ = .23. Qualtrics sample model fit: AIC = 306; $R^2_{McF}$ = .47.

betting and at-risk/problem gambling in this sample was significant, $\chi2(1, N = 580) = 21.20$, $p < .001$. A binary logistic regression analysis for the Qualtrics sample showed that at-risk/problem gamblers were significantly more likely to report both esports cash betting and esports skin betting in the last four weeks, compared to non-problem gamblers (Table 6).

## The relationship between esports cash betting and esports skin betting and at-risk/ problem gambling, controlling for other gambling activities

Hierarchical logistic regressions to test Objective 4 revealed that betting on esports with skins significantly predicted at-risk/problem gambling, even after participation in other, traditional

**Table 5. Multivariate logistic regression results with past-month participation in monetary gambling activities as predictors of past-month esports skin betting.**

| | Advertisements Sample Esports skin betting vs not (ref. = not) | | | | | Qualtrics Sample Esports skin betting vs not (ref. = not) | | | | |
|---|---|---|---|---|---|---|---|---|---|---|
| Variables | B | S.E. | Z | p | Odds Ratio [95% CI] | B | S.E. | Z | p | Odds Ratio [95% CI] |
| EGMs | .30 | .19 | 1.65 | .099 | 1.36 [.94,1.95] | .63 | .41 | 1.56 | .119 | 1.88 [.85,4.16] |
| Racing | 2.03 | .42 | 4.83 | < .001 | 7.64 [3.35,17.44] | .60 | .40 | 1.50 | .133 | 1.82 [.83,3.97] |
| Scratchies, lotto, pools | .70 | .18 | 3.86 | < .001 | 2.01 [1.41,2.86] | 1.54 | .26 | 5.93 | < .001 | 4.65 [2.80,7.73] |
| Keno | - | - | - | - | - | .26 | .41 | .51 | .614 | 1.23 [.55,2.72] |
| Bingo or housie | .82 | .18 | 4.48 | < .001 | 2.28 [1.59,3.27] | .71 | .35 | 2.03 | .042 | 2.03 [1.03,4.03] |
| Poker | - | - | - | - | - | -1.01 | .53 | -1.91 | .056 | .36 [.13,1.03] |
| Casino games | - | - | - | - | - | .71 | .51 | 1.37 | .169 | 2.03 [.74,5.55] |
| Sports | - | - | - | - | - | -.17 | .43 | -.40 | .691 | .84 [.36,1.96] |
| Fantasy sports | .27 | .19 | 1.41 | .159 | 1.32 [.90,1.93] | .64 | .42 | 1.53 | .126 | 1.90 [.84,4.34] |
| Informal private betting | .49 | .18 | 2.64 | .008 | 1.63 [1.13,2.33] | .67 | .30 | 2.20 | .028 | 1.95 [1.08,3.51] |
| Intercept | -2.01 | .15 | -13.68 | < .001 | .13 [.10,.18] | -2.83 | .17 | -17.00 | < .001 | .06 [.04,.08] |

Missing data, indicated by dashed lines, are due to insufficient sample size for analyses ($n < 10$). *EGMs* = electronic gaming machines.

The reference group for all predictor variables = No (did not participate in the past month).

Advertisements sample model fit: AIC = 873; $R^2_{McF}$ = .10. Qualtrics sample model fit: AIC = 530; $R^2_{McF}$ = .24.

**Table 6. Logistic regression results with past-month esports cash and skin betting participation as predictors of at-risk/problem gambling.**

| Variables | Qualtrics sample At-risk/problem gambler (ref. = non-problem gambler) | | | | |
|---|---|---|---|---|---|
| | *B* | S.E. | Z | *p* | Odds Ratio [95% CI] |
| Esports cash betting | .92 | .30 | 3.10 | .002 | 2.50 [1.40,4.47] |
| Esports skin betting | 1.66 | .28 | 5.89 | < .001 | 5.25 [3.02,9.11] |
| Intercept | -.54 | .13 | -4.30 | < .001 | .58 [.46,.75] |

Reference group = NO for predictor variables, Esports cash betting and Esports skin betting. Insufficient sample size for analyses of Advertisements sample data, as $n < 10$ for non-problem esports cash bettors: See text for analysis of esports skin betting data for this sample.

forms of gambling were controlled for (Table 7). This was consistent across both samples. In the Qualtrics sample, past-month esports cash gambling was not significantly associated with at-risk/problem gambling after controlling for these other gambling forms (as noted above, this could not be examined in the Advertisements sample due to low sample size).

Follow-up hierarchical logistic regressions to address revealed that, in the Advertisements sample, recent esports skin bettors were over 3 times more likely to be at-risk/problem gamblers, even after controlling for the total number of past-month monetary gambling forms they participated in (Table 8).

In the Qualtrics sample, esports cash bettors were not more likely to be at-risk/problem gamblers, after controlling for the total number of gambling forms participated in during the past month. However, esports skin bettors were over 3 times more likely to be at-risk/problem gamblers, after controlling for both esports cash betting and the total number of gambling forms participated in during the past month.

**Table 7. Multivariate hierarchical logistic regression results with different past-month gambling activities as predictors of at-risk/problem gambling.**

| Variables | Advertisements Sample At-risk/problem gambler (ref. = non-problem gambler) | | | | | Qualtrics Sample At-risk/problem gambler (ref. = non-problem gambler) | | | | |
|---|---|---|---|---|---|---|---|---|---|---|
| | *B* | S.E. | Z | *p* | Odds Ratio [95% CI] | *B* | S.E. | Z | *p* | Odds Ratio [95% CI] |
| EGMs | - | - | - | - | - | .95 | .44 | 2.19 | .029 | 2.59 [1.11,6.08] |
| Racing | - | - | - | - | - | .48 | .41 | 1.20 | .232 | 1.62 [.73,3.58] |
| Scratchies/lottery/pools | 1.60 | .34 | 4.69 | < .001 | 4.93 [2.53,9.60] | .07 | .26 | .27 | .784 | 1.07 [.65,1.77] |
| Keno | - | - | - | - | - | .26 | .44 | .59 | .554 | 1.30 [.55,3.08] |
| Bingo | - | - | - | - | - | .40 | .33 | 1.23 | .219 | 1.50 [.79,2.84] |
| Sports | - | - | - | - | - | .56 | .41 | 1.39 | .165 | 1.76 [.79,3.88] |
| Fantasy sports | - | - | - | - | - | s.57 | .42 | 1.38 | .167 | 1.78 [.79,4.05] |
| Informal private betting | 2.15 | .30 | 7.27 | < .001 | 8.59 [4.81,15.33] | .08 | .27 | .30 | .762 | 1.09 [.64,1.85] |
| Esports cash betting | - | - | - | - | - | .07 | .37 | .20 | .844 | 1.08 [.52,2.22] |
| Esports skin betting | 1.35 | .33 | 4.05 | < .001 | 3.86 [2.01,7.42] | 1.34 | .30 | 4.56 | < .001 | 3.83 [2.12,6.91] |
| Constant | .05 | .18 | .31 | .760 | 1.06 [.75,1.49] | -.83 | .16 | -5.20 | < .001 | .44 [.32,.60] |

Missing data, indicated by dashed lines, are due to insufficient sample size for analyses($n < 10$). *EGMs* = electronic gaming machines. For the outcome variable, At-risk/Problem gambling = Non-problem gambler ($n = 203$) vs at risk/problem gamblers ($n = 204$). Reference groups for the predictor variables, each gambling activity = No participation in the past month. Model 1for both samples = Other gambling activities; Model 2 for Advertisements sample = Esports skin betting added. Model 2 for Qualtrics sample = Esports cash betting added and Model 3 = Esports skin betting added. Advertisements sample model fit: Model 1 AIC = 413, $R^2_{McF}$ = .20, Likelihood $\chi^2(2) = 101.00$, $p < .001$; Model 2 AIC = 395; $R^2_{McF}$ = .24, Likelihood $\chi^2(3) = 120.00$, $p < .001$. There was a significant improvement from Model 1 to 2, $\chi^2(1) = 19.70$, $p < .001$. Qualtrics sample model fit: Model 1 AIC = 506, $R^2_{McF}$ = .14, Likelihood $\chi^2(7) = 76.10$, $p < .001$; Model 2 AIC = 508; $R^2_{McF}$ = .14, Likelihood $\chi^2(8) = 76.50$, $p < .001$; Model 3 AIC = 488; $R^2_{McF}$ = .17, Likelihood $\chi^2(9) = 97.90$, $p < .001$. There was no significant improvement from Model 1 to 2, $\chi^2(1) = .45$, $p = .502$. Model 3 was a significantly better fit for the data than Model 2, $\chi^2(1) = 21.41$, $p < .001$.

**Table 8. Multivariate hierarchical logistic regression results with total number of past-month gambling activities as predictors of at-risk/problem gambling.**

| | Advertisements sample | | | | | Qualtrics sample | | | | |
| | At-risk/problem gambler (ref. = non-problem gambler) | | | | | At-risk/problem gambler (ref. = non-problem gambler) | | | | |
| **Variables** | **B** | **S.E.** | **Z** | **p** | **Odds Ratio [95% CI]** | **B** | **S.E.** | **Z** | **p** | **Odds Ratio [95% CI]** |
| --- | --- | --- | --- | --- | --- | --- | --- | --- | --- | --- |
| Number of Other Gambling Activities | 2.39 | .24 | 10.13 | < .001 | 10.89 [6.86,17.28] | .39 | .07 | 5.19 | < .001 | 1.47 [1.27,1.71] |
| Esports cash betting | - | - | - | - | - | -.01 | .36 | -.03 | .974 | .99 [0.49,1.98] |
| Esports skin betting | 1.14 | .44 | 2.58 | .010 | 3.13 [1.32,7.43] | 1.31 | .30 | 4.40 | < .001 | 3.69 [2.06,6.60] |
| Intercept | -2.29 | .34 | -6.62 | < .001 | .10 [.05,.20] | -.94 | .15 | -6.23 | < .001 | .39 [.29,.53] |

Missing data, indicated by dashed lines, are due to insufficient sample size for analyses ($n < 10$). For the outcome variable, At-risk/Problem gambling = Non-problem gambler ($n = 203$) vs at risk/problem gamblers ($n = 204$). Reference groups for the predictor variables, each gambling activity = No participation in the past month. Model 1for both samples = Total Number of Other Gambling Activities (1 to 10); Model 2 for Advertisements sample = Esports skin betting added. Model 2 for Qualtrics sample = Esports cash betting added and Model 3 = Esports skin betting added.

Advertisements sample model fit: Model 1 AIC = 212, $R^2_{McF}$ = .59, Likelihood $\chi^2(1)$ = 299.00, $p < .001$; Model 2 AIC = 207; $R^2_{McF}$ = .60, Likelihood $\chi^2(2)$ = 306.00, $p < .001$. There was a significant improvement from Model 1 to 2, $\chi^2(1)$ = 7.28, $p = .007$.

Qualtrics sample model fit: Model 1 AIC = 491, $R^2_{McF}$ = .14, Likelihood $\chi^2(1)$ = 76.80, $p < .001$; Model 2 AIC = 493; $R^2_{McF}$ = .14, Likelihood $\chi^2(2)$ = 76.90, $p < .001$; Model 3 AIC = 474; $R^2_{McF}$ = .17, Likelihood $\chi^2(3)$ = 97.80, $p < .001$. There was no significant improvement from Model 1 to 2, $\chi^2(1)$ = .15, $p = .701$. Model 3 was a significantly better fit for the data than Model 2, $\chi^2(1)$ = 20.83.41, $p < .001$.

## Discussion

This study examined the characteristics of adolescent esports bettors, and relationships between their esports betting, video gaming activities, monetary gambling participation, and at-risk/problem gambling status. Both esports cash betting and esports skin betting were included. Two samples with different recruitment methods were analysed to enhance confidence in the results that were consistent across both samples. The discussion below focuses most on these consistent results.

Both samples showed no differences between males and females, or with their parents' living situation, amongst adolescents who had bet on esports with either cash or skins in the last month, compared to those who had not. Gender differences in video gaming participation have largely disappeared amongst youth [36] and this may also apply to the linked activity of esports betting. In contrast, a UK study of emerging adults aged 16–24 years found that past-year esports c ash bettors were more likely to be male [23]. The current study compared past-month participation in esports betting between genders, rather than past-year participation, which may explain the different result found. Both of our samples also showed no significant difference by age in past-month esports cash betting, consistent with Wardle et al.'s [23] finding for past-year participation. However, our two samples yielded inconsistent results for age and esports skin gambling. Aboriginal and/or Torres Strait Islander participants were more likely to report participating in esports cash betting in both samples, but in esports skin betting only in one sample.

The psychological factors of impulsivity and wellbeing showed inconsistent results across the samples and for the two types of esports betting (cash and skins). These mixed results may indicate that these psychological factors are not strong drivers of adolescent esports betting, even though impulsivity and lower wellbeing are typically strong risk factors for adult gambling involvement [37,38]. Perhaps the higher level of impulsivity generally found amongst adolescents, compared to adults, accounts for its lesser explanatory value in this younger age group.

In alignment with Macey and Hamari's findings [18,21], recent esports bettors who used cash or skins were more likely to engage in esports gaming activities, compared to those who had not recently bet on these activities. This was most apparent amongst esports skin bettors who, in both samples, were more likely to have recently played, watched, and competed in

esports. Amongst esports cash bettors, watching esports competitions and playing an esports video game were each only significant in one sample. These results suggest that adolescent esports skin bettors are more likely to engage in a broader range of esports-related activities, but causal relationships remain unclear. Playing, watching, and competing in esports activities may encourage young people to bet on esports, especially because skins are an easier way for underage people to bet compared to using cash; alternatively, betting on esports may encourage them to take more interest in these games. Further, esports skin bettors may be particularly drawn to playing esports video games for the chance to win more skins for their esports betting. This heightened play on esports video games may also contribute to problematic gaming symptoms, as found amongst esports skin bettors in one sample, since video games have structural characteristics that can contribute to excessive and persistent play [39–42].

The current study also examined participation in other gambling activities by the adolescent esports bettors. Recent cash and skin esports bettors were more likely to have recently gambled on lottery-type games, bingo and informal private betting, across both samples. These are amongst the most common monetary gambling activities that adolescents engage in [11,22,43,44], so it is not surprising that esports bettors tended to engage in these activities. In both samples, recent fantasy sports betting was also more common amongst esports cash bettors, which may reflect a broader interest in digital sports amongst this group.

A significant positive relationship was found in both samples between at-risk/problem gambling and past-month engagement in esports skin betting, as well as with past-month engagement in esports cash betting in the sample with sufficient respondents in the relevant subgroups for this analysis. When controlling for other types of gambling, as well as the total number of gambling activities engaged in over the past month, at-risk/problem gamblers were not more likely to have recently bet on esports with cash. However, in both samples, at-risk/problem gamblers were more likely to have participated in esports skin betting, even when controlling for recent esports cash gambling. Overall, these results indicate that esports skin gambling is uniquely associated with at-risk/problem gambling amongst youth, but this relationship is not apparent for esports cash betting. This finding is consistent with Greer et al. [13], who found that esports skin betting, but not esports cash betting, was a unique risk factor for greater problem gambling severity amongst adults.

## Implications

The results from this study highlight that underage betting on esports is a public health concern, particularly if bets are placed using skins. Esports are increasingly popular, especially amongst young people [3,6]. Further, COVID-19 restrictions have fuelled a surge in esports betting to replace gambling activities that have been unavailable, and catalysed industry development to acquire alternative revenue streams [45,46]. These trends suggest that underage esports betting, including the use of skins, is likely to continue growing, and it is already an issue of regulatory and community concern [7,9,47].

While only a small minority of adolescents currently engages in esports cash betting [11,22,23], esports skin betting appears to be much more popular in this age group (11). This may be attributable to the popularity of skins and easy access to unregulated skin betting websites. Esports skin gambling may be particularly harmful to adolescents, and/or appeal to those at-risk of harm since they obtain the currency for betting through games of chance in video games and then use this currency to participate in further betting. Given that gambling problems in adolescents were uniquely predicted by esports skin gambling, our results support calls for improved regulation and monitoring of esports betting and skin gambling operators, as well as rigorous age verification processes to prevent access by minors. Young people would

also benefit from education about the risks of underage gambling and of betting with unregulated esports betting and skin gambling operators, as well as the marketing tactics used by industry operators. Parents can be supported through resources that increase their awareness and knowledge of esports betting and gambling with skins, so they are better equipped to protect their children from gambling harm.

## Limitations, strengths and future research

While this study was limited by its use of non-probability samples, its purposive sampling meant we could recruit adequate respondents in sub-groups of interest to enable most of the planned analyses. Further, analysing two samples enhanced confidence in the results, which were reasonably consistent in most analyses, despite the different recruitment method for each sample. However, further research in different samples is needed to assess the generalisability of the results. Naturally, the study was also limited by its cross-sectional design and self-report data which may be subject to recall and social desirability biases, although asking about past-month betting should have helped to reduce some recall bias. Due to legislation prohibiting underage gambling, low numbers of participants reported participating in some gambling activities, especially in the Advertisements sample. There were even fewer participants who reported gambling but did not experience problems. While providing important insights into underage gambling, these factors mean that analyses exploring the relationship between participation and at-risk/problem gambling should be subject to further verification and replication. In addition, given that the psychological factors examined had no consistent relationship with esports betting participation, future research could examine a wider range of variables. Environmental factors such as parental and peer influences, and situational variables, such as convenient access, may be more salient drivers of esports betting participation amongst young people. While these results appear to be consistent with previous research [18,21,23], further inquiry is necessary to test this assumption, as well as to explore how these factors influence transitions to harmful and problem gambling later in life. Nevertheless, the current study provides valuable, targeted insight into the risk factors that correlate with esports cash betting and esports skins betting, highlighting the need for these gambling activities (especially the latter) to be appropriately regulated to help reduce associated problems and harm.

## Conclusion

This study has provided new information about betting on esports by adolescents who are under the legal gambling age. This is an issue that has attracted community concern, but little empirical research. Our findings indicate that the individual factors examined, including demographics, impulsivity and wellbeing, had surprisingly little bearing on adolescent engagement in recent esports betting. Rather, engagement in esports gaming activities, such as playing and watching esports, was associated with heightened participation in esports betting. Importantly, esports betting using skins uniquely predicted gambling problems amongst the adolescents, even after controlling for other gambling activities that are competing sources of gambling harm. Easy access to a product that currently lacks effective age-gating, is often provided by unlicensed operators, and whose recent and regular use predicts gambling problems, is a dangerous combination for underage adolescents and warrants improved efforts to provide protection for this vulnerable population.

## Author Contributions

**Conceptualization:** Nerilee Hing, Alex M. T. Russell, Matthew Rockloff, Daniel L. King, Matthew Browne.

**Data curation:** Alex M. T. Russell.

**Formal analysis:** Lisa Lole.

**Funding acquisition:** Nerilee Hing, Alex M. T. Russell, Matthew Rockloff, Daniel L. King, Matthew Browne.

**Investigation:** Nerilee Hing, Alex M. T. Russell.

**Methodology:** Nerilee Hing, Lisa Lole, Alex M. T. Russell.

**Project administration:** Nerilee Hing.

**Supervision:** Nerilee Hing.

**Writing – original draft:** Nerilee Hing, Lisa Lole.

**Writing – review & editing:** Lisa Lole, Alex M. T. Russell, Matthew Rockloff, Daniel L. King, Matthew Browne, Philip Newall, Nancy Greer.

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
