## [Decision Letter · Decision Letter 0]

27 Jan 2022

PONE-D-21-39632Adolescent betting on esports using cash and skins: Links with gaming, monetary gambling, and problematic gamblingPLOS ONE

Dear Dr. Hing,

Thank you for submitting your manuscript to PLOS ONE. After careful consideration, we feel that it has merit but does not fully meet PLOS ONE’s publication criteria as it currently stands. Therefore, we invite you to submit a revised version of the manuscript that addresses the points raised during the review process.

We look forward to receiving your revised manuscript.

Kind regards,

Marc Potenza

Academic Editor

PLOS ONE

Journal Requirements:

2. Peer review at PLOS ONE is not double-blinded (https://journals.plos.org/plosone/s/editorial-and-peer-review-process). For this reason, authors should include in the revised manuscript all the information removed for blind review.

"Funding for this study was provided by the NSW Government’s Responsible Gambling Fund, with support from the NSW Office of Responsible Gambling."

We note that you have provided funding information. However, funding information should not appear in the Acknowledgments section or other areas of your manuscript. We will only publish funding information present in the Funding Statement section of the online submission form. 

"NH, AR, MR, MB, PN, NG and DK received grant funding to conduct this research (no grant number assigned). Funding for this study was provided by the NSW Government’s Responsible Gambling Fund, with support from the NSW Office of Responsible Gambling https://www.responsiblegambling.nsw.gov.au. The funders reviewed the survey instrument for this study but had no role in data collection, analysis, decision to publish, or preparation of the manuscript."

Reviewers' comments:

Reviewer's Responses to Questions

**Comments to the Author**

1. Is the manuscript technically sound, and do the data support the conclusions?

Reviewer #1: Yes

Reviewer #2: Yes

2. Has the statistical analysis been performed appropriately and rigorously? 

Reviewer #1: Yes

Reviewer #2: No

3. Have the authors made all data underlying the findings in their manuscript fully available?

Reviewer #1: No

Reviewer #2: Yes

4. Is the manuscript presented in an intelligible fashion and written in standard English?

Reviewer #1: Yes

Reviewer #2: Yes

5. Review Comments to the Author

Reviewer #1: 1.The introduction is well written. Regarding the objectives, the authors mentioned that one of their objectives is to ’examine rates of participation in other forms of gambling among past-month esports cash bettors and esports skin bettors.’ However, it seems to me that the objective is not directly reached in the study. Besides, I am also wondering if the convenience sampling design of the study can serve the objective very well.

2.The study recruited two samples of Australians aged 12-17 years. I am wondering what is the reasoning/considerations of choosing adolescents aged 12-17 years old?

3.The authors conducted the analyses by the classification of different recruitment methods. I think that separate analyses with two samples are okay but the classification used in the study is questionable. Because of the nature of convenience sampling, significant differences between the samples on key variables are common. As there may be many other options of grouping which could be applied, grouping by recruitment methods may not be the best one for cross-validation in the study. Personally, I think the author needs to elaborate their considerations on the choice of the classification in the manuscript.

4.In the PROCEDURE section, the authors mention an university human research ethics committee approval number ‘ which is XXXX. It is unusual, please confirm if it is correct.

5.Please fix a typo ‘）’ in line 386.

Reviewer #2: Overall, the manuscript is nicely written and the quality of the tables is satisfactory. The reference list covers the relevant literature adequately and in an unbiased manner (not more than 15 % self-citation). The study adheres to ethical standards including ethics committee approval and consent procedure.

I have a few minor comments with regards to before recommending to publish.

1) Authors should indicate the study’s design in the title or the abstract.

2) The scientific background and rationale for the investigation being reported is explained well. However, having this knowledge should result in formulation of prespecified hypotheses. Otherwise, this reads as an exploratory study which is not the case.

3) Authors should specify what they mean by ‘location’ eligibility criteria.

4) Where available, provide Cronbach alpha of the questionnaires used in both samples.

5) Please include statistical analysis part with clearly defined outcomes, predictors and potential confounders. Describe how the confounding variables were decided, were they based on the preliminary analysis or they used from the literature? Detail the reasons of using/not using covariates in each of the regression models. In other words make clear which confounders were adjusted for and why they were included. Explain how the variables were entered in the regression models.

6) Explain how the required study size was arrived at.

7) In the legend of Table 1, please describe questionnaires used to obtain results.

8) Please describe the engagement rate for the Qualtrics study. Is it consistent with other studies?

9) In the results part first paragraph I would like to suggest summarizing key results with reference to the study objectives.

10) Consider providing strengths of the work, not only limitations.

11) Discuss the external validity of the study results.

6. PLOS authors have the option to publish the peer review history of their article (what does this mean?). If published, this will include your full peer review and any attached files.

Reviewer #1: **Yes: **Jiang Long

Reviewer #2: No

---

## [Author Response · Author response to Decision Letter 0]

8 Mar 2022

Dear Professor Potenza

Below we explain the adjustments we have made to adhere to the journal’s requirements, and our responses to the reviewers’ comments. We thank the reviewers for their helpful comments which have guided us in improving this manuscript.

Kind regards

Nerilee

Journal Requirements:

Response: The manuscript has been revised to make it compliant with PLOS ONE’s style requirements.

2. Peer review at PLOS ONE is not double-blinded (https://journals.plos.org/plosone/s/editorial-and-peer-review-process). For this reason, authors should include in the revised manuscript all the information removed for blind review.

Response: We have now included information that was blinded for review in the original manuscript.

"Funding for this study was provided by the NSW Government’s Responsible Gambling Fund, with support from the NSW Office of Responsible Gambling."

We note that you have provided funding information. However, funding information should not appear in the Acknowledgments section or other areas of your manuscript. We will only publish funding information present in the Funding Statement section of the online submission form. 

"NH, AR, MR, MB, PN, NG and DK received grant funding to conduct this research (no grant number assigned). Funding for this study was provided by the NSW Government’s Responsible Gambling Fund, with support from the NSW Office of Responsible Gambling https://www.responsiblegambling.nsw.gov.au. The funders reviewed the survey instrument for this study but had no role in data collection, analysis, decision to publish, or preparation of the manuscript."

Response: The funding information has been removed from the Acknowledgements section. We do not wish to make any amendment to the original funding statement provided (this has been noted in the cover letter).

Response: The Procedure section of the Method in the revised manuscript now explained: “The study procedures were carried out in accordance with the Declaration of Helsinki. The study’s protocol was approved by the Central Queensland University Human Research Ethics Committee. All subjects were informed about the study, and all provided written informed consent. Parental consent was also sought for all participants.”

Response: Changes to the reference list include the addition of 5 additional articles, including: 

[12] Williams RJ, Volberg RA, Stevens RM. The population prevalence of problem gambling: Methodological influences, standardized rates, jurisdictional differences, and worldwide trends. 2012 Ontario Problem Gambling Research Centre.

[14] Kim HS, Wohl MJ, Gupta R, Derevensky JL. Why do young adults gamble online? A qualitative study of motivations to transition from social casino games to online gambling. Asian J Gambl Issues Public Health. 2017 7(1):1-11.

[15] King DL, Delfabbro PH. Adolescents’ perceptions of parental influences on commercial and simulated gambling activities. Int Gambl Stud. 2016 16:424-441.

[16] King D L, Delfabbro PH. The convergence of gambling and monetised gaming activities. Curr Opin Behav Sci. 2020 31:32-36.

[35] Faul F, Erdfelder E, Buchner A, Lang AG. Statistical power analyses using G*Power 3.1: Tests for correlation and regression analyses. Behav Res Meth 2009 41:1149-1160.

 

Response to reviewers’ comments

Reviewer #1: 

1.The introduction is well written. Regarding the objectives, the authors mentioned that one of their objectives is to ’examine rates of participation in other forms of gambling among past-month esports cash bettors and esports skin bettors.’ However, it seems to me that the objective is not directly reached in the study. Besides, I am also wondering if the convenience sampling design of the study can serve the objective very well.

Response: The words ‘rates of’ has been deleted to better reflect the aims of the study and the use of a convenience sample (please see Page 6). 

2.The study recruited two samples of Australians aged 12-17 years. I am wondering what is the reasoning/considerations of choosing adolescents aged 12-17 years old?

Response: The current study sought to explore gambling in teenagers, before they turn the legal gambling age (18 years) in Australia. Adolescence is a critical stage of development that is qualitatively different from both childhood and adulthood, in terms of esports and skins gambling behaviours, but knowledge of this phenomenon is lacking in the existing literature. A brief rationale for this has been added to Page 4 of the revised manuscript.

3.The authors conducted the analyses by the classification of different recruitment methods. I think that separate analyses with two samples are okay but the classification used in the study is questionable. Because of the nature of convenience sampling, significant differences between the samples on key variables are common. As there may be many other options of grouping which could be applied, grouping by recruitment methods may not be the best one for cross-validation in the study. Personally, I think the author needs to elaborate their considerations on the choice of the classification in the manuscript.

Response: The classification of the 2 samples (according to recruitment method) was decided upon, due to differences observed for recruitment methods in previous studies and discrepancies observed in preliminary analyses (e.g., Hing et al., 2021; 11). Showing the results from both samples strengthens the results that are consistent between samples by demonstrating they are not just an artefact of a particular recruitment method. Rationale for this has been included in the newly-added ‘Statistical Analyses’ section (Pages 11-13).

4.In the PROCEDURE section, the authors mention an university human research ethics committee approval number ‘which is XXXX. It is unusual, please confirm if it is correct.

Response: The ‘no. XXXX’ text in the procedure section has been deleted.

5.Please fix a typo ‘）’ in line 386.

Response: This typo has been corrected.

Reviewer #2: 

Overall, the manuscript is nicely written and the quality of the tables is satisfactory. The reference list covers the relevant literature adequately and in an unbiased manner (not more than 15 % self-citation). The study adheres to ethical standards including ethics committee approval and consent procedure. I have a few minor comments with regards to before recommending to publish.

1) Authors should indicate the study’s design in the title or the abstract.

Response: The study’s design (descriptive cross-sectional) has been indicated in the abstract.

2) The scientific background and rationale for the investigation being reported is explained well. However, having this knowledge should result in formulation of prespecified hypotheses. Otherwise, this reads as an exploratory study which is not the case.

Response: Due to the lack of research into adolescent gambling on esports, and the descriptive cross-sectional nature of the research, the research team considered that it was not appropriate to provide specific hypotheses for the study.

3) Authors should specify what they mean by ‘location’ eligibility criteria.

Response: Location eligibility criteria means that participants lived one specific Australian state, New South Wales (NSW). This has been clarified on Page 7 of the revised text.

4) Where available, provide Cronbach alpha of the questionnaires used in both samples.

Response: The Cronbach alpha of the questionnaires, for both samples, has been added in the Materials section of the revised manuscript (please see Pages 9 and 10).

5) Please include statistical analysis part with clearly defined outcomes, predictors and potential confounders. Describe how the confounding variables were decided, were they based on the preliminary analysis or they used from the literature? Detail the reasons of using/not using covariates in each of the regression models. In other words make clear which confounders were adjusted for and why they were included. Explain how the variables were entered in the regression models.

Response: A separate section called Statistical Analyses has been added to the manuscript. This section outlines the predictor and outcome variables for each analysis (these were previously included in the results section, alongside the corresponding analyses, but have been relocated to this section and more details added; please see Pages 11-13 of the revised manuscript). 

6) Explain how the required study size was arrived at.

Response: The required sample size was determined by a power analysis (in the program G*Power) and feasibility estimates provided by the panel provider. This detail has been added to the ‘Statistical Analyses’ section (Page 11).

7) In the legend of Table 1, please describe questionnaires used to obtain results.

Response: The questionnaires used to obtain the results in Table 1 have been noted in the legend/notes (Page 8).

8) Please describe the engagement rate for the Qualtrics study. Is it consistent with other studies?

Response: By engagement rate, we believe that the reviewer is referring to the number of people who start the survey based on the number of people who are contacted. This information usually cannot be known because, unlike a telephone survey, it is usually unclear how many people are contacted by the panel provider. Qualtrics draws from several different participant panels, and these panels work in different ways to each other: some send invitations via email, some via a web portal or app, and others post links on a “feed” within an app, amongst other approaches. While some of these methods might be trackable, such as the number of emails sent, many panels use a combination of these approaches, making it difficult to determine the number of people invited. Further, some panels target invitations based on the profiles of potential respondents, and some do not. For some studies, Qualtrics may draw from more than one panel (with appropriate deduplication checks to ensure that people who are on more than one panel are not able to complete the survey more than once). Because of these differences across panels, it is usually not possible to know how many people were invited. This also means that there is no general engagement rate for Qualtrics studies, because it depends on the individual panels that partner with Qualtrics at that point in time. As such, we cannot report the engagement rate for this project. However, all efforts have been made to be transparent about completion rates, including full details about exclusions and incomplete responses.

9) In the results part first paragraph I would like to suggest summarizing key results with reference to the study objectives.

Response: The key objective and results of each analysis have been summarised in each paragraph of the results (Pages 13, 19, 25, and 27).

10) Consider providing strengths of the work, not only limitations.

Response: The strengths of the work, including the targeted sampling strategy employed in the research, have been provided in the discussion section (please see Page 37).

11) Discuss the external validity of the study results.

Response: We have now noted that further research in different samples is needed to assess the generalisability of the results. The finding that environmental factors and situational variables may be more salient drivers of esports betting participation amongst young people appears to be consistent with other preliminary research findings. However, due to the lack of rigorous research into this specific topic, the external validity of these results can only be presumed and needs to be subjected to future research. This has been acknowledged in the revised manuscript (Page 37).

---

## [Editor Report · Decision Letter 1]

23 Mar 2022

Adolescent betting on esports using cash and skins: Links with gaming, monetary gambling, and problematic gambling

PONE-D-21-39632R1

Dear Dr. Hing,

We’re pleased to inform you that your manuscript has been judged scientifically suitable for publication and will be formally accepted for publication once it meets all outstanding technical requirements.

Kind regards,

Marc Potenza

Academic Editor

PLOS ONE
---

## [Editor Report · Acceptance letter]

27 Apr 2022

PONE-D-21-39632R1 

Adolescent betting on esports using cash and skins: Links with gaming, monetary gambling, and problematic gambling 

Dear Dr. Hing:

I'm pleased to inform you that your manuscript has been deemed suitable for publication in PLOS ONE. Congratulations! Your manuscript is now with our production department. 

Kind regards, 

on behalf of

Dr. Marc Potenza 

Academic Editor

PLOS ONE